# Influence of Scoring Systems on Mental Fatigue, Physical Demands, and Tactical Behavior during Soccer Large-Sided Games

**DOI:** 10.3390/ijerph20032087

**Published:** 2023-01-23

**Authors:** Jesús Díaz-García, José Carlos Ponce-Bordón, Abel Moreno-Gil, Ana Rubio-Morales, Miguel Ángel López-Gajardo, Tomás García-Calvo

**Affiliations:** Faculty of Sport Sciences, University of Extremadura, 10003 Cáceres, Spain

**Keywords:** cognitive exertion, football, soccer constraints, tactical behavior, training load

## Abstract

Constraints are common in soccer training to develop physical, technical-tactical, and mental training concurrently. This study examined how different scoring systems influence physical, tactical, and mental demands during large-sided games in soccer. Eighteen youth-elite male (17.39 ± 1.04 y) soccer players completed three 8 vs. 8 large-sided games where the different score systems were i) official score system (OSS; i.e., 1 goal = 1 goal), ii) double the value of the goal—4 min (DVx4; i.e., 1 goal = 1 goal from 0.00 to 7.59 min, and 1 goal = 2 goals from 8.00 to 12.00 min), and iii) double the value of the goal—8 min (DVx8; i.e., 1 goal = 1 goal from 0.00 to 3.59 min, and 1 goal = 2 goals from 4.00 to 12.00 min). Physical demands and tactical behaviors were recorded during tasks using a global positioning system and video camera. Mental fatigue was recorded pre- and post-task using a visual analogue scale. Also, the ratio of perceived exertion and mental load were recorded after tasks were finished. Results reported the highest values of mental and physical demands in DVx4. Mental fatigue increased during all three large-sided games, although this increase was significantly higher in DVx4 compared with OSS (*p* = 0.006) and DVx8 (*p* = 0.027). Tactical behavior showed a trend towards more direct play during DVx4, which was less observed during DVx8, and not at all during OSS. In conclusion, changing the scoring system affects physical, tactical, and mental demands.

## 1. Introduction

Soccer is a mainly endurance-based sport interspersed with repeated high-intensity efforts [1] that lead to physical fatigue among soccer players [2]. However, soccer is mentally as well as physically fatiguing [3,4]. Soccer is perceptual-cognitive: players must remain alert for extended periods, constantly analyzing a dynamic environment and selecting only the relevant information [5]. Previous studies have reported negative effects of physical [6] and mental fatigue [7,8,9] on soccer performance. For that reason, experts have become interested in the different physical, technical-tactical, and mental demands of different soccer training methodologies [10]. This knowledge about training demands may be helpful for soccer coaches to optimize players’ training load and performance [11].

Small, medium, or large-sided games (SSG, MSG, or LSG, respectively) are common training methods in soccer to replicate the physical [12,13], technical-tactical [14,15] and mental demands [16,17,18] of competition. The manipulation of constraints during SSGs, MSGs, or LSGs allows coaches to modify the physical [19,20], technical-tactical [14,21], and/or mental demands of the tasks [22,23] according to their specific training purposes. Compared to more traditional training methods (i.e., extensive repetition of physical or technical skills out of soccer-specific context), these exercises (i.e., SSG, MSG, and LSG) are perceived as more soccer-specific and allow for optimization of training time, since physical, technical-tactical and mental skills are developed concurrently [24].

Previous studies have analyzed the effects of different constraints on the physical, technical-tactical, and mental demands of soccer players during training. The use of constraints depends on their effects on fatigue (e.g., coaches should not use high-fatiguing constraints close to competitions), the point of the training day in the microcycle (e.g., there should be a progressive reduction or tapering in the load across the microcycle) and the point in the season (e.g., there should be less use of high-fatiguing constraints during periods of important matches) [25]. The effects of pitch size, the number of players, and the ratio of the number of players to pitch area have been widely analyzed. Notably, it has been reported that larger pitch size and higher pitch area per player are associated with higher total distances covered, more high-intensity distances covered, and easy maintenance of ball possession [20,26,27]. Another widely analyzed constraint is the change of rules or configurations during SSGs. Studies show that constraints that cause a higher entropy (referring to the uncertainty of the task [28]) among players are related to higher mental effort [17,22] and more significant difficulties in maintaining the fluency of the game, which is associated with less time playing and less physical effort [15,29]. The effects of other constraints, such as as coaches’ behavior [15,29,30] or player unbalance [21] have also been analyzed. The authors have concluded that dynamic behavior of coaches results in more physical and mental effort from the players. Meanwhile, player unbalance increases high-intensity efforts and facilitates the fluency of the game due to a reduction in the ratio of players to pitch area.

There are many constraints whose effects should be analyzed. The present study focuses on how the scoring system impacts the physical, mental, and technical-tactical responses of soccer players during LSGs. This constraint has been examined because it is widely used during soccer training; coaches sometimes modify the standard scoring systems, establishing a double value for specific goals. The present study aimed to compare the effects of the normal scoring system with the effects of a modified scoring system applied for two different times: a) double (x2) value of the goals scored within the final 4 min of the task, and b) double (x2) value of the goals scored within the final 8 min of the task. A previous study has suggested that increasing the value of the goals in the final part of the task increases the mental load and fatigue in SSG [17]. However, this study did not consider LSG or tactical behavior. Another study has suggested that the score of the matches may increase the activity profile and the skill attempts of players, but it has not been tested if this happens during LSG [31]. Based on previous studies [17,31], we hypothesized that the changes in the scoring system would increase the mental demands and mental fatigue of the players (Hypothesis 1), as well as their physical demands (Hypothesis 2), and would cause changes in the technical-tactical behaviors of the players (Hypothesis 3). 

## 2. Materials and Methods

### 2.1. Sample

A total of 18 youth-elite male (M_age_ = 17.39 ± SD_age_ = 1.04) soccer players that competed in the First U18 Spanish National Division participated in the study. Players had an average experience of 9.31 ± 2.51 years playing soccer, and the team trained 4 days per week. All players (or their parents, in the case of U18 players) signed an informed consent before the start of the study. Players were encouraged to avoid consuming caffeine 1 h before the training, and creatine during the study, due to their ergogenic effects on mental fatigue. 

### 2.2. Instruments and Outcomes

#### 2.2.1. Polar Team Pro (Polar Electro, Finland, 2015) 

To measure the physical demands during the training tasks, this global positioning system (GPS) was used. Specifically, the outcomes were i) internal load variables: mean and peak heart rate; ii) external load variables: mean and peak speed and distance/minute covered. This technology uses a concentration of signals from different Polar brand sensors, which was designed for the control of physical activity in collective sports like soccer. Indeed, it is currently one of the most used instruments for this purpose in soccer [32]. The validity of the GPS system was previously reported [33].

#### 2.2.2. Ratio of Perceived Exertion 

The ratio of perceived exertion (RPE) was used to quantify the perception of physical effort from players using the CR-10 scale. This was asked as: How much physical effort did the task require? The range of responses included values from 0 (not at all tired) to 10 (maximum level of perceived exhaustion). The use and accuracy of the RPE for this purpose has been previously proven in soccer [34]. The RPE was recorded immediately after the LSG had finished.

#### 2.2.3. NASA Task Load Index

To quantify the mental load perceived by soccer players, an adaptation of the NASA task load index (NASA TLX) was used. This adaptation asked about six specific subscales of the NASA TLX: i) mental effort (i.e., how much mental effort was required during the task?); ii) physical effort (i.e., how much physical effort was required during the task?); iii) time pressure (i.e., what time pressure did you feel due to the pace of the task?); iv) performance satisfaction (i.e., how satisfied are you with your performance during the task?); v) general effort (i.e., how much general effort was required?); and vi) unsafety (i.e., how unsafe did you feel?). The range of responses included values from 0 (no effort perceived) to 10 (maximum effort perceived) for each item described. Previous studies have already used this instrument to quantify mental load in soccer [35]. The NASA TLX was recorded immediately after the LSG had finished.

#### 2.2.4. Visual Analogue Scale

The visual analogue scale 100 (VAS100) was used to assess the subjective perception of mental fatigue by players. Participants were instructed to mark their current perceived state of mental fatigue on a line from 0 (no mental fatigue perceived) to 100 (maximum mental fatigue perceived). The accuracy of this scale has been previously reported in athletes‘ ratings of mental fatigue [36,37]; the scale has been used in soccer studies [38], and it has been described as the most sensitive instrument for measuring mental fatigue in athletes [39]. The VAS was recorded immediately before the LSG started and just after the LSG had finished.

#### 2.2.5. Video Camera HDR-PJ30VE (Sony, Tokyo, Japan)

To record the tactical behavior of the players, the researchers filmed the LSG tasks using a video camera. Tactical behaviors of the players were analyzed, taking into account: zone of the pitch where possession was recovered (defensive and offensive pitch), zone of the pitch where possession was lost (defensive and offensive pitch), number of passes (defensive and offensive pitch), possession time (defensive and offensive pitch), number of shots (directly out, corner, save without corner, and goal), and zone of the shots (inside and outside the area). 

### 2.3. Experimental Design

All research procedures were conducted in accordance with the Declaration of Helsinki and had the approval of the Ethics Committee for Research with Human Beings (approval number: 93/2020). 

The study was an experimental design performed on one real competitive soccer team using an ecological design. Three different LSG situations (8 vs. 8 + goalkeepers) over an artificial grass turf of 70 × 40 meters were performed for 12 min each. There were 48 h of recovery between LSG exercises (no competitive match was performed between them). Only the scoring system changed between LSG exercises. The three experimental conditions were: (i) official scoring system (OSS; i.e., 1 goal = 1 goal for all time during the task); (ii) double value of the goal in the last four minutes (DVx4; i.e., 1 goal = 1 goal from 0.00 to 7.59 min, and 1 goal = 2 goals from 8.00 to 12.00 min); (iii) double value of the goal in the last eight minutes (DVx8; i.e., 1 goal = 1 goal from 0.00 to 3.59 min, and 1 goal = 2 goals from 4.00 to 12.00 min). The order of the sessions was randomly selected to guarantee the ecological design of the study: so, DVx8, OSS, and DVx4 were performed firstly, secondly, and thirdly, respectively.

A familiarization session was held before the study to guarantee that all the soccer players understood the questionnaires provided. The schedule of the experimental sessions was: (i) 8-min warm-up (i.e., low-intensity running and mobility exercises, managed by the same researcher for all players); (ii) researchers put the GPS on the players; (iii) pre-VAS was recorded; (iv) LSG and video recording started at the same time; (v) immediately after the LSG exercises, video recording finished, and post-VAS, RPE and NASA TLX were recorded. 

### 2.4. Statistical Analysis

The analyses were performed using the Statistical Package for the Social Sciences (SPSS), version 25.0. Data were presented as means ± standard deviation (SD). The Shapiro–Wilk test was used to check the normality of the data. Sphericity was verified by Mauchly’s test. When the assumption of sphericity was not met, the significance of F ratios was adjusted with the Greenhouse–Geisser procedure. If the data were not normally distributed, the Wilcoxon test was used to check possible differences between experimental sessions on mental effort (i.e., NASA TXL), physical effort (i.e., variables from GPS) and perceptions (i.e., RPE), and tactical behavior. The Wilcoxon test was also used to check possible changes in subjective mental fatigue (i.e., VAS) from pre- to post-experimental sessions. A ∆ mental fatigue (subjective mental fatigue post-session - subjective mental fatigue pre-session) was also calculated. Possible differences in the ∆ mental fatigue of the different experimental protocols were compared using the Kruskall–Wallis test. Statistical significance for all analyses was set at *p* < 0.05, *p* < 0.01, and *p* < 0.001.

## 3. Results

The comparison between different LSG situations for mental load and mental fatigue is shown in Table 1. With regard to mental load, DVx4 was identified by players as the most mentally (*p* < 0.001 and *p* = 0.008 when compared with OSS and DVx8, respectively), physically (*p* = 0.009 and *p* = 0.032 when compared with OSS and DVx8, respectively), temporally (*p* < 0.001 and *p* = 0.042 when compared with OSS and DVx8, respectively) and generally (*p* < 0.001 and *p* = 0.006 when compared with OSS and DVx8, respectively) demanding situation. DVx8 was also identified as more mentally (*p* = 0.008), physically (*p* = 0.026), temporally (*p* < 0.001) and generally (*p* = 0.034) demanding than OSS. However, performance satisfaction was significantly higher in OSS than in DVx4 (*p* = 0.046). There were no significant differences in this variable between OSS and DVx8 (*p* = 0.069), nor between DVx4 and DVx8 (*p* = 0.078). Meanwhile, unsafety was significantly lower in OSS than in DVx4 (*p* = 0.039), without significant differences in this variable between OSS and DVx8 (*p* = 0.061), nor betweenDVx4 and DVx8 (*p* = 0.034). With regard to mental fatigue, all the situations caused a significant increase in mental fatigue. Significant changes were observed between pre- and post-mental fatigue in OSS (*p* = 0.009), DVx4 (*p* < 0.001) and DVx8 (*p* < 0.001). In addition, DVx4 caused significant higher increases from pre- to post-experimental mental fatigue than DVx8 (*p* = 0.006) and OSS (*p* = 0.027), meanwhile, DVx8 showed significantly higher increases from pre- to post-experimental protocol in this variable than OSS (*p* = 0.041).

The physical effort performed by soccer players during LSG situations are shown in Table 2. DVx4 showed the highest values of Mean Heart Rate (with significant differences when compared with OSS (*p* = 0.042)), Peak Heart Rate (with significant differences when compared with OSS (*p* = 0.044)), RPE (with significant differences when compared with OSS (*p* = 0.009) and DVx8 (*p* = 0.031)), Distance/Minute (with significant differences when compared with OSS (*p* = 0.037)), and Mean Speed (with significant differences when compared with OSS (*p* = 0.049)). In all these variables, values of DVx8 were higher than those observed in OSS (with significant differences in Peak Heart Rate (*p* = 0.042) and Peak Speed (*p* = 0.027)). DVx8 showed the highest values of Peak Speed (with significant differences when compared with OSS (*p* = 0.019)). In this variable, the values of DVx4 were significantly higher than those of OSS (*p* = 0.025).

The tactical behaviors of the players for each LSG situation are shown in Table 3. With regard to the zone of ball recovery, higher number of recoveries in the defensive pitch were observed during OSS (*p* = 0.009 when compared with DVx4; and *p* = 0.032 when compared with DVx8). On the contrary, the results also showed a trend to recover more balls in the offensive pitch in DVx4 and DVx8 when compared with OSS (*p* < 0.001 and *p* = 0.007, respectively). With regard to the zone of the pitch where possession was lost, DVx4 showed a significant higher number of ball possession losses when compared with OSS (*p* = 0.033) and DVx8 (*p* = 0.005). However, a significantly higher number of ball possession losses happened in the offensive pitch in OSS when compared with DVx4 (*p* = 0.023) and DVx8 (*p* < 0.001). Indeed, significant differences were observed in this variable between DVx4 and DVx8 (*p* = 0.006), with a higher value in DVx4. With regard to the number of passes, during DVx4, players performed a significantly smaller number of passes both in the defensive (*p* < 0.001 when compared both with OSS and DVx8) and offensive (*p* < 0.001 when compared both with OSS and DVx8) phases, in comparison with the other protocols. This was similarly reflected in the possession time. During DVx4, there weresignificantly smaller possession times both in the offensive (*p* < 0.001 when compared both with OSS and DVx8) and defensive phases (*p* < 0.001 when compared both with OSS and DVx8), in comparison with the other situations. Indeed, DVx8 also showed a significantly smaller possession time in the offensive phase when compared with OSS (*p* < 0.001). With regard to the zone of the shots, during DVx8, players showed a significantly higher trend to shots out of area compared with DVx4 (*p* < 0.001) and OSS (*p* < 0.001).

## 4. Discussion

The present study aimed to compare the effects of three different scoring systems on the physical and mental demands of, and tactical behaviors performed by, soccer players during LSG. The main findings of this study were that the three scoring systems resulted in different physical and mental demands and tactical behaviors. DVx4 and DVx8 resulted in higher physical demands and were more mentally demanding and fatiguing than OSS. The increases in physical and mental effort, and tactical changes, in DVx4 were particularly significant, compared with DVx8. The analysis of tactical behavior showed a trend towards more direct play during DVx4 and DVx8, compared with OSS. 

Firstly, we hypothesized that adaptations in the scoring system would increase the players’ mental demands and mental fatigue (*Hypothesis 1*). Players identified DVx4 and DVx8 as more mentally demanding and fatiguing than OSS. Notably, DVx4 was reported as more significantly mentally demanding and fatiguing by soccer players. Therefore, Hypothesis 1 can be confirmed. The three scoring systems used in the study resulted in significant increases in the mental fatigue reported by players after all of the different LSG situations. This is in line with previous evidence suggesting that soccer is a mentally fatiguing activity [2,3,4]. The cognitive (e.g., concentration or decision making) and emotional (e.g., anxiety) processes involved in soccer seem to cause mental fatigue among players [38]. The higher impact of the double value of the goals in the final score than the official system (i.e., 1 goal = 1 goal) could explain the presence of higher feelings of entropy (i.e., uncertainty) among players, as well as the associated higher mental demands and fatigue observed. The higher values of mental demands and mental fatigue reported by players during DVx4 than during DVx8 (although the impact of goals on these games were the same) may be explained by the temporal pressure of the tasks [18]. The authors explain that players feel more mental demand and fatigue when the temporal pressure increases. Higher temporal pressure is related to stress, anxiety, and impulse control related to higher levels of mental fatigue [18]. In this case, players had more time in DVx8 than in DVx4 to counteract the impact that a double goal caused on the final score. 

Secondly, we also hypothesized that changes in the scoring system would increase the physical demands of the players during LSG exercises (*Hypothesis 2*). LSG influenced the technical-tactical demands of the game. Then, we also hypothesized that modifications in the scoring system would cause changes in the technical-tactical behavior of the players (*Hypothesis 3*). The results of the present study suggest that the modifications performed in the scoring systems significantly increased the game’s physical demands and caused a trend towards more direct play during DVx4 and DVx8, with respect to OSS. Therefore, Hypotheses 2 and 3 can be also confirmed. Previous studies have suggested that the score of a soccer game influences the physical and technical-tactical demands of the game. Specifically, time playing while losing seems to reduce a team’s time of possession due to the teams’ betting on a more direct style of play [40]. This is in line with the results of the present study, where players spent less time in possession and performed a smaller number of passes, both in the defensive and offensive pitch, when the impact of a goal in the score were higher (i.e., DVx4 and DVx8). Similarly to what was observed with the mental demands, the effects of this constraint on physical and technical-tactical demands were higher during DVx4 than during DVx8. As we previously explained, the higher temporal pressure of this task enhances the mentioned effects of this modification to the scoring system compared with DVx8. Then, these adaptations cause a change in the tactical behavior of the teams, which try to play more directly. This change in the tactical behavior impacts on physical effort as well, due to the presence of more space (i.e., more meters between lines of players) caused by the change in the style of play to allow players to cover more distance and perform a higher number of high-intensity efforts [40].

## 5. Conclusions

The present study confirms that coaches can change soccer players’ mental and physical demands and technical-tactical behaviors, by changing the scoring system of the LSG. This study showed how an increase in the value of the goals in the final part of the tasks (i.e., 1 goal = 2 goals) significantly increased the mental and physical demands and mental fatigue of the tasks; this was accompanied by a more direct style of play compared with the traditional scoring system (i.e., 1 goal = 1 goal). These increases in efforts and changes in the style of play were further enhanced when this modification in the scoring system was applied for the last four minutes than when it was applied for the last eight minutes. This situation seems to be caused by higher temporal pressure when the adaptation was applied for a shorter time.

## Figures and Tables

**Table 1 ijerph-20-02087-t001:** Mental load and fatigue reported by players. A comparison between experimental sessions.

Variables	OSS	DVx4	DVx8	Between Experimental Session Comparison
Mental Effort	6.44 ± 2.44	8.36 ± 2.26	7.22 ± 3.12	a ***, b **, c **
Physical Effort	4.39 ± 1.19	5.19 ± 1.88	4.79 ± 2.24	a **, b *, c *
Temporal Pressure	3.66 ± 1.87	5.91 ± 2.04	5.42 ± 1.39	a ***, b ***, c *
Performance Satisfaction	6.87 ± 2.22	6.56 ± 2.01	6.71 ± 3.11	a *
General Effort	5.31 ± 1.16	6.12 ± 1.92	5.52 ± 2.03	a ***, b *, c **
Unsafety	6.04 ± 1.72	6.31 ± 1.99	6.12 ± 1.32	a *
Mental fatigue	Pre Post	Pre Post	Pre Post	a **, b *, c *
	5.25 5.32	5.19 5.19	5.32 8.29 ***	

* Note. Data was presented as means ± SD. *** *p* < 0.001; ** *p* < 0.01; * *p* < 0.05. a = significant differences between OSS and DVx4; b = significant differences between OSS and DVx8; c = significant differences between DVx4 and DVx8.

**Table 2 ijerph-20-02087-t002:** Physical efforts covered and reported by players. A comparison between experimental sessions.

Variables	OSS	DVx4	DVx8	Between Experimental Session Comparison
Mean Heart Rate	162.31 ± 15.18	169.71 ± 16.21	165.34 ± 18.33	a *
Peak Heart Rate	184.31 ± 6.12	189.66 ± 4.88	188.91 ± 5.12	a *, b *
Ratio of Perceived Exertion	5.01 ± 1.09	5.77 ± 1.87	5.29 ± 1.16	a **, c *
Distance/Minute	103.37 ± 8.99	108.34 ± 9.16	106.34 ± 9.31	a *
Peak Speed	19.76 ± 1.96	22.68 ± 3.55	23.12 ± 2.21	a *, b *
Mean Speed	10.65 ± 3.56	11.31 ± 4.39	11.04 ± 3.55	a *

* Note. Data was presented as means ± SD. ** *p* < 0.01; * *p* < 0.05. a = significant differences between OSS and DVx4; b = significant differences between OSS and DVx8; c = significant differences between DVx4 and DVx8.

**Table 3 ijerph-20-02087-t003:** Analysis of tactical behavior during the large-sided games. A comparison between experimental sessions.

Variables		OSS	DVx4	DVx8	Between Experimental Session Comparison
Zone of the pitch where possession was recovered	Defensive pitch	8.50	7.50	7.00	a *, b **
	Offensive pitch	2.00	3.50	3.00	a ***, b **
Zone of the pitch where possession was lost	Defensive pitch	2.00	2.50	1.00	b *, c **
	Offensive pitch	8.25	7.50	6.00	a *, b ***, c **
Number of passes	Defensive pitch	31.50	21.00	28.50	a ***, c ***
	Offensive pitch	20.00	11.00	21.00	a ***, c ***
Possession time (s)	Defensive pitch	98.00	76.50	95.50	a ***, c ***
	Offensive pitch	70.75	26.50	57.50	a ***, b ***, c ***
Shoots	Out (i.e., Goal kick)	1.25	1.00	1.50	
	Corner (i.e., goalkeeper or defender throw the ball out)	0.75	1.50	0.50	a **, c **
	Save (blockage or similar that did not produce corner)	0.50	0.00	1.00	
Zone of the shoots	Inside the area	1.50	1.00	1.00	
	Outside the area	1.75	2.00	3.50	b ***, c ***

* Note. Data was presented as means ± SD. *** *p* < 0.001; ** *p* < 0.01; * *p* < 0.05. a = significant differences between OSS and DVx4; b = significant differences between OSS and DVx8; c = DVx4 and DVx8.

## Data Availability

Data is unavailable due to privacy or ethical restrictions.

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
