# Peer review of "Influence of Scoring Systems on Mental Fatigue, Physical Demands, and Tactical Behavior during Soccer Large-Sided Games"

_ijerph, 2023, doi:10.3390/ijerph20032087_

Round 1

Reviewer 1 Report

The research presented in this article was scientifically well conceptualized and executed. However, some areas require attention to bring them to a level of publication. The authors should attend to the following to edit in pdf.

Author Response

Thanks for your work! We have uploaded a specific letter of responses and a new version of the manuscript with track changes.

Reviewer 2 Report

- Why you don't use the same p value to compare the Kruskall-Wallis test?

- I recomened to use the same format of decimals in Table 3.

Author Response

Thanks for your work! we have uploaded a specific letter of responses and a new version of the manuscript with track changes.
